# Challenges and Emerging Trends in Toner Waste Recycling: A Review

Meera Parthasarathy

Sri Ranga Ramanuja Centre for Advanced Research in Sciences, Department of Chemistry, Srimad Andavan Arts & Science College, Tiruchirappalli 620005, India; meerasarathy07@gmail.com

**Abstract:** Toner waste is one of the major electronic waste materials posing serious environmental threat and health hazards. Globally, only about 20–30% of toner waste is recycled, while the remaining percentage is dumped in landfills. Recycling options are limited due to the desirably engineered durability of toners, ascribed to a complicated composition of chemicals, carbon black, and plastic particles, which in turn creates critical challenges in recycling. The World Health Organization has classified toner waste as class 2B carcinogen due to its potential health hazard. In this review, the existing challenges in toner waste recycling are discussed from the perspective of environmental, health, and feasibility aspects. In parallel, the challenges have been opening up alternative strategies to recycle toner wastes. Emerging trends in toner waste recycling include transformation of toner waste into value-added products, utilization as raw material for nanomaterial synthesis, generation of composite electrodes for power generation/storage devices, integration into construction materials, and development of microwave absorbing composites. Considering the enormous volume of toner waste generated globally every year, better recycling and transformation strategies are needed immediately. A circular economy could be established in the future by transforming the enormous toner waste into a resource for other applications. For an effective management of toner waste in the future, an integrated approach involving policies and legislations, infrastructure for collection and treatment, and financial planning among the stakeholders is needed in addition to technological innovations.

**Keywords:** toner waste; copier; air pollution; remanufacturing; extended producer responsibility; cartridge recycling

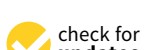

## 1. Introduction

The demand for electrical and electronic equipment (EEE) has been increasing exponentially, due to urbanization, industrialization, and increasing population. Temperature exchange equipment (refrigerators, air conditioners), computer monitors, lamps, large electrical appliances (washing machines, dishwashers, electric stoves), small equipment (vacuum cleaners, microwave ovens, ventilators, etc.), and IT/telecommunication equipment (printers, personal computers, telephones, etc.) are the different categories of EEE. As a critical player in the global economy, EEE has equally become inevitable to our daily life. The waste stream generated by EEE consists of harmful as well as valuable components and is collectively termed "e-waste". As per the global e-waste monitor, 53.6 Mt e-waste was generated in the year 2019 out of which only 17.4% was collected and recycled through formal means [1]. Printers and cartridges constitute a significant source of e-waste despite the modern trend of digitization. About one million printer cartridges are disposed every day on a global scale. Each cartridge contains about eight percent of unused toner by weight, amounting to the release of 6000 tons of carbon powder into the environment. Other hazardous materials released from unused and residual toner powder include plastics, heavy metals, and carcinogens such as polycyclic aromatic hydrocarbons and resins [2,3]. Further, the small particle size, which becomes smaller after use due to dust particles in the residual toner powder causes air pollution and respiratory problems [4].

Global e-waste statistics in 2020 shows that developed countries generate more e-waste per capita than developing countries (Figure 1) [1]. Moreover, the percentage of e-waste collected and recycled through the formal sector is higher in developed countries. However, the actual percentage of recycling might be higher than calculated due to possible lapses in documentation of e-waste collection and recycling in the formal sector. A number of regulatory policies and legislations are being enforced to make the practice of e-waste collection, documentation, and recycling systematic. Extended Producer Responsibility (EPR) has been a major driver for e-waste recycling especially in developed countries [5]. The EPR policy holds the producers responsible for collecting and recycling the end-of-life products. Moreover, the economics of collection and recycling are to be borne by the manufacturers who might include the costs in the market price of products. In fact, EPR covers a broader perspective holding the manufacturer responsible throughout the lifecycle of the product. The policy is intended to implement resource efficiency and sustainable development at the product design level. However, in the present scenario and due to issues in managing the enormous amount of e-waste, it is being applied mainly to the end-of-life products.

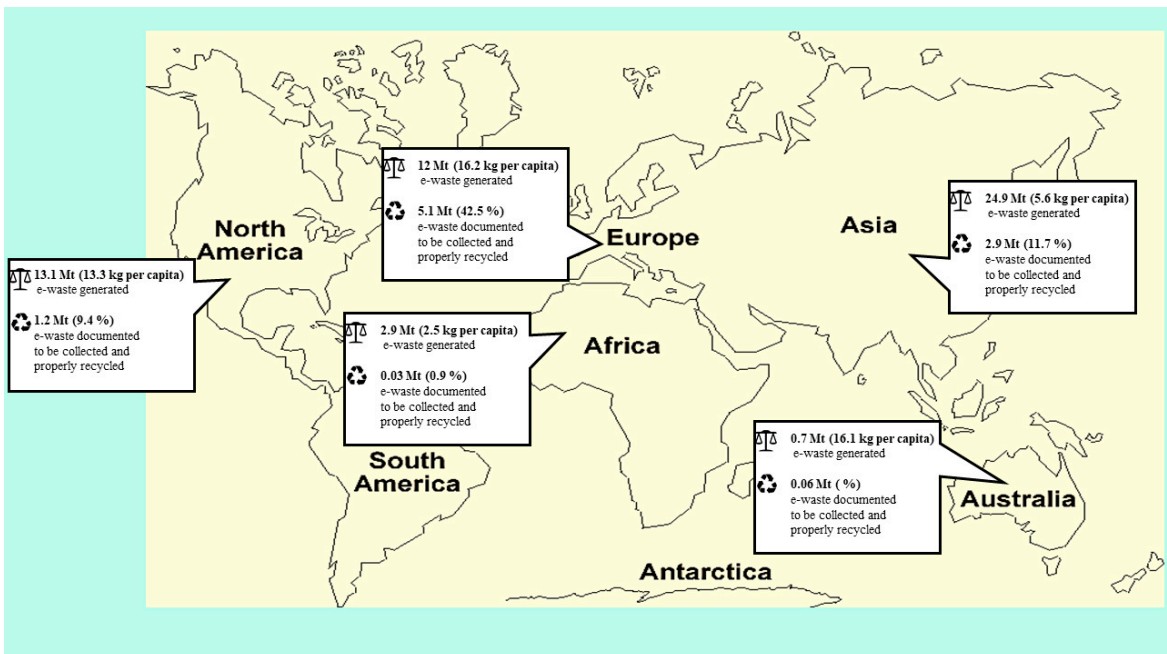

**Figure 1.** Global statistics of e-waste generation and formal recycling in the year 2019 (adopted with permission from Ref. [1]).

Despite the legislation based on EPR policies, developed countries export a significant amount of e-waste to developing countries. The reasons are mainly the cost and labour needed for the formal recycling of e-waste. The Secure E-waste Export and Recycling Act (SEERA) insists on domestic recycling of untested nonworking electronics and stopped the export of these items to developing countries. However, the act does not control the export of scrap material and functional devices. The European Environment Agency estimates that nearly 1.3 million tons of e-waste is exported annually to continents such as Africa and Asia [6]. About 23,000 metric tons of e-waste was exported illegally from the United Kingdom to India, Africa, and Asia in 2003 [6]. The United States, on the other hand, was reported to contribute eight percent of its e-waste to the "global hidden flow", which includes illegal export, dumping in landfills, and incineration [6]. Such illegal transport of e-waste results in informal recycling practices and related environmental/health issues in developing countries [7]. Implementation of EPR policy is even more challenging in developing countries, which lack a sufficient infrastructure, governance, and economy [6].

Printer toners constitute an important part of electronic waste, mainly due to their limited operational life, resistance to degradation after disposal, and environmental and economic challenges in recycling/reuse. When disposed in landfills, they cause soil and water pollution leading to a multitude of health hazards [8]. Negligence in the treatment, recycling, and reuse of toner waste is expected to create serious environmental issues in the near future. Considering the current global scenario and future issues, it is important to develop viable technologies for recycling and reuse of toner waste. In this context, the present review provides a comprehensive account of the challenges and emerging trends in the recycling and reuse of toner waste.

## 2. Composition of Toners

Inkjet printers use printing inks, which contain chemicals such as butyl urea, cyclohexanone, ethoxylated acetylenic diols, ethylene diamine tetra-acetic acid, ethylene glycol, and several sulfur-containing dyes. During the printing process, many volatile organic solvents are released. Most of the components in printing inks are harmful chemicals, which pollute the environment [9]. Simple household inkjet printers contain two cartridges—one for black and the other for cyan, magenta, and yellow. If one of the colours is completed, the second cartridge has to be replaced altogether. The same is true in the case of high-end inkjet printers containing multiple cartridges for better colour reproduction in flex and photographs. Frequent replacement of multiple cartridges on depletion of even one colour results in bulk disposal of cartridges, thereby causing serious environmental pollution.

Laser printers and copiers use toners, either in the dry form or wet form. Dry toners consist of acrylic and styrene powders together with colour pigments. Liquid toners consist of acrylic resins with added dye pigments to provide bright colour images. Laser printers are available in monocomponent and dual component models. In the former, toner powder and magnetic particles are mixed in a single printer drum. In the latter model, these are kept in separate drums and mixed with a developer while printing. Plastic resins constitute a major portion (45–90%) of toner powder and are generally made of styrene and acrylic polymers [3]. Magnetic properties are imparted to the toner through iron oxide ingredients. In addition, various metals and semiconductors are added to the toner powder to induce triboelectric and superflow properties [10]. Such a combination of organic and inorganic materials makes the toner powder stable and steadily fixes it to papers. Nevertheless, these ingredients render the toner nearly nondegradable and eco-unfriendly, posing serious environmental threats. Moreover, the small particle size (~10 μm) leads to serious respiratory problems and lung damage [11,12].

## 3. Challenges in Recycling Toner Waste

The maximum utilization of toners in xerographic processes is 90% and the remaining 10% of unused toner is collected in a waste bin. Around 66% of the unused toner in the waste bin is used in toner remanufacturing, thus reducing the use of virgin raw materials. The remaining (34%) toner waste consists of other impurities such as paper bits and staples, which are first removed by a preliminary screening process. The efficiency of screening process is 95% and the toner can be readily introduced into the remanufacturing process [13]. With the increasing amounts of toner waste dumped into landfills, concerns about recycling and reusing are raising gradually. Considering electronic waste as a whole, the proportion of toner waste was not considered as significant. However, there is a growing concern lately due to the increasing health issues related to air and water quality. The three major drivers for e-waste recycling are environmental concerns, energy/resource conservation, and economics [14]. These could be useful drivers for toner recycling as well. When unused and waste toners are dumped into the landfill, they pollute air and ground water easily due to their fine particle size (8–10 μm). With the threatening respiratory pandemic, COVID-19, it is important to maintain a hygienic pollution-free environment. In a recent study, the risk of hospitalization of COVID-19 positive patients was found to be higher when exposed to

particulate emissions [15]. Even the risk of transmission of COVID-19 is higher in areas with air polluted with particulate emissions [16].

The major challenges in recycling toner waste are:

- Lack of awareness of the toxic effect of waste toners among the end users and those involved in informal recycling;
- Inaccessible and insufficient collection points for systematic collection of waste toners;
- Inherent stability of toners, which makes recycling/reusing difficult;
- Lack of strict legislation and governance in e-waste management, especially in developing countries;
- Uncontrolled growth of informal recycling sector;
- Pollution arising from irresponsible processing of waste toners.

The foremost challenge is the collection of waste toners from end users from different geographical locations. Collection and transportation might be more challenging in the case of waste toners compared to bulk electronic waste due to the fine particle size. Individual awareness is needed for the responsible collection of waste toners from xerographic and printing devices. In the case of waste electrical and electronic equipment (WEEEs), local waste pickers collect and transport them for informal recycling. In developing countries, this exercise has been preventing dumping of electronic waste in trash. However, compared to the market for WEEEs, toner waste has not reached much attention as a valuable resource for informal recycling. As a result, almost all of the toner waste is being disposed as solid waste and dumped in landfills. It is also difficult to monitor and quantify irresponsible dumping of toner waste, especially in populated countries. EPR policy has an excellent scope and can really serve as a driving force for the management of toner waste. Legislations based on the EPR policy are already in place in developed countries, and they do have the option of implementing them. When it comes to developing countries, there are a number of challenges in implementing EPR and related legislations [17]. EPR-based legislations for e-waste management have been recently introduced in some developing countries (Table 1). Still, in the case of toner waste management, the following challenges exist in the implementation of EPR policies.

**Table 1.** EPR-based legislations in developing countries [18].

| Country | Specific or Draft Legislation |
|---------|-------------------------------|
| China | Rules on the administration of the recovery and disposal of discarded electrical and electronic products (promulgated in 2009, effective in 2011) |
| India | E-waste management and handling rules (2016) |
| Indonesia | Specific article on EPR is under preparation under Solid Waste Management Act 2008 |
| Malaysia | Specific article on take-back and deposit refund in Solid Waste and Public Cleansing Management Act 2007. Draft regulation on recycling and disposal of End-of-Life electrical and electronic equipment. |
| Thailand | WEEE Strategic Plan 2007 and Draft Act on Economic Instruments for Environmental Management (under development) |
| Vietnam | Draft regulations on the reclamation and treatment processes for disposal products (under planning; draft released in 2010) |

- It is impossible to identify the producer of a particular toner at the end of use.
- Waste toners need to be collected at accessible locations. Toners of different compositions will be collected together. Unified recycling methods are needed to treat the mixture. In this case, the cost and responsibility of recycling has to be borne by the government and the producer cannot be held responsible. Common treatment/recycling plants, such as ETPs (Effluent Treatment Plants) for treating industrial effluents, need to be commissioned by local governments.

- Local xerographic and printing shops might not use branded toners for their devices.
- There is no mechanism to monitor, quantify, and document the usage and disposal of toners.
- Information labels on recyclability are not available on toner products.

The processibility of toners is another important challenge in recycling [19]. Toners are designed to be stable, chemically and mechanically, through a complicated chemical composition (Table 2). The plastic particles in toners are meant to withstand the varying temperature, pressure, and humidity conditions of printers. The international standard for permanence and durability, ISO 11798, demands printer toners to be light fastening, water-, heat-, and wear-resistant. While the stability of the toners is necessary to obtain a high-quality print, it is a serious challenge to recycling. The process of recycling involves mechanical dismantling and separation of the components, followed by density separation. In the later stages, recycling is completed using hydrothermal and various metallurgical processes [20]. As the toner powders are resistant to water, heat, and mechanical forces, recycling using conventional methods becomes difficult. Thermal recycling of waste toner might also be dangerous due to possibilities of explosion [21]. The composition of toners also affects the deinking efficiency of office waste papers and paper recycling post printing [22]. At the same time, the whole recycling process becomes less economic. Due to the economic challenges in recycling toner waste, even developed countries are struggling to adhere to the formal recycling norms laid by the government [14,23].

**Table 2.** Composition of toners [24].

| Component | Percentage (%) |
| --- | --- |
| Polymers | 45 |
| $Fe_3O_4$ | 25 |
| Carbon black | 20 |
| Additives | 5 |
| Wax | 3 |
| Cellulose/Kaolin | 1 |
| Surfactants | 1 |

Instead, many developed countries have been exporting their e-waste to developing countries, despite stringent regulations including the Basel Ban (1989 Basel Convention on the control of transboundary movements of hazardous wastes and their disposal) and SEERA (2012). The Basel Ban controls only hazardous waste and SEERA controls only untested electronics. So, the exporters are cleverly labelling their e-waste and scrap materials as nonhazardous waste and secondhand electronics and sending them to developing countries. The developing countries, on the other hand, willingly import e-waste from the developed countries due to the economic benefits of informal recycling [25–27]. A major portion of e-waste in developing countries is recycled by the informal sector [25]. Most of the operations of informal recycling are performed manually in a less-organized manner without any regulations for operational safety. The formal and informal operations of e-waste recycling in China are compared in Figure 2. Informal recycling is conducted by poor families for their livelihood. Low wages, market demand for secondhand electronic products, and lack of environmental regulations allow them to earn a viable profit margin in the informal recycling of e-waste.

A major downside of informal recycling practice is the environmental pollution and onsite as well as end-of-pipe health hazards caused due to the byproducts [28]. Toner dust causes serious respiratory problems to the workers involved in the recycling process [29]. Mathias et al. reported that toner particles caused ROS-induced oxidative stress on A549 lung cells and might be genotoxic with frequent exposure [30]. Heavy metals present in toner powder are found to accumulate in the environment and human population around the recycling workshops. A study conducted in Guiyu, a major e-waste recycling town in China indicated elevated blood levels of lead in children living near the recycling

workshops [28]. Cadmium content in printer toners and inks is known to damage kidney and bones [31]. Toner powder cannot be disposed of in landfills or incinerated due to the release of noxious gases such as dioxin and furan under high temperature and pressure [3]. Despite these issues, the informal recycling sector has become as a major player. Hence, it would be beneficial to formalize or integrate the informal sector with the formal recycling sector. It is noteworthy that the definitions of formal and informal recycling are not always used in the stringent sense but might vary from country to country [25]. A recent review of e-waste policy issues in developing countries concludes that sustainable management of e-waste is possible only if formal and informal recycling are integrated [32].

## 4. Emerging Trends in Recycling Toner Waste

The common 3R strategies of waste management, 'reduce, reuse, and recycle', are applicable to the management of toner waste as well [33]. With the evolving demands for printing, the chances of reducing the usage of toners might be scarce. Nonetheless, a number of avenues are opening up for the other two options, reusing and recycling of toner waste. Even in modern printers, around 13% of toner powder is rejected as waste in the cleaning sump.

Some manufacturers have attempted to divert the waste toner back to the fresh toner reservoir of the xerographic machine. But the attempts were in vain, as the waste toner does not adhere to the paper due to lack of fusible particles [34]. Reuse of waste toner is nearly impossible in the case of mixed colour toners. To solve this problem, individual colours were first separated from the mixed colour waste toner using charge-based separation techniques. The separation techniques are yet to reach technical success in separating all the component colours from the mixed colour waste toners. An alternative approach is 'remanufacturing', in which reclaimed waste toner is processed further for optimal reuse. A few companies such as Canon and HP have launched the practice of remanufactured printer cartridges. Under this scheme, old printer cartridges are collected, shredded, pelletized, and new cartridges are moulded from the reclaimed pellets [35]. Toner powder has been remanufactured by mixing waste toner with carbon black nanoparticles and a small proportion of unused fresh toner. The remanufactured toner powder, named as 'hybrid black toner' produced good print quality, providing a promising route to the management of toner waste [34]. Remanufactured toner cartridges are found to be about 65% cheaper by cost than the OEM cartridges, and the print quality is also comparably good [36].

Recent attempts are focused on the generation of value-added products from waste toner, without polluting the environment (Table 3). Ruan et al. have developed a set of processes for the treatment of waste toner and integrated in the production line of waste printer cartridge recycling system [24]. Using this technology, microplastics and $Fe_3O_4$ nanoparticles were recovered from waste toners in a series of eco-friendly vacuum gasification and condensation processes (Figure 3). In addition to preventing environmental pollution, the technology recovers valuable products from waste toner. Quite recently, Gaikwad et al. recovered pure iron (98%) from waste toner through an environmentally-friendly thermal transformation process [37]. Uttam Kumar and coworkers reported an innovative method of using plastics from end-of-life printers as a carbon source for the reduction in iron oxide in waste toners to produce iron [38]. This approach thus simultaneously abates the pollution issues of waste toner as well as those of e-waste plastics. Waste toner from colour printers has also found application as filler and colorant in the manufacture of rubber products [8]. Ruan et al. reported the use of waste toners to produce synthetic oils [39]. Parthasarathy and coworkers found an interesting application of waste toner powder in the treatment of domestic waste water [40]. The hybrid composition of waste toner including organic/microplastics and inorganic ($SiO_2$ and $Fe_3O_4$ nanoparticles) enables the removal of a wide range of organic and inorganic pollutants from sewage water. The waste toner recovered after water treatment was hydrophilic with clay-like consistency and hence became processable for other applications.

**Table 3.** Properties of toner waste and emerging applications.

| Property of Waste Toner | Emerging Applications | Reference |
|---|---|---|
| Composition | Recovery of pure iron and iron oxides, nanomaterial synthesis | [9,37,39,41] |
| Viscosity | Filler for rubbers | [10] |
| Adsorptive capacity | Wastewater treatment | [40,41] |
| Electrical conductivity | Conducting polymer blends | [42] |
| Charge storage capacity | Power generation/storage devices | [43,44] |
| Mechanical strength | Building construction | [45–48] |
| Thermal conductivity | Microwave absorbing composites | [49] |

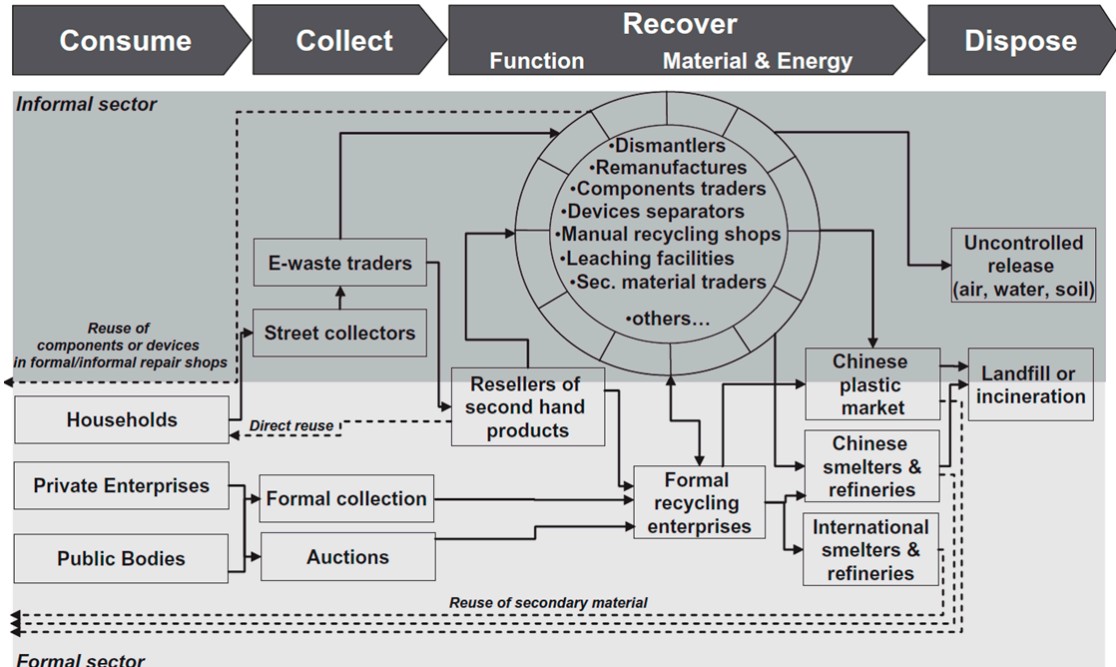

**Figure 2.** Commonly adopted e-waste recycling methods in the formal and informal sectors in China (reproduced with permission from Ref. [25]).

An interesting approach towards toner recycling is the utilization of waste toner in the synthesis of nanomaterials for different applications. Toner powder itself is an organic–inorganic hybrid nanocomposite material. A major portion (~55%) of toner powder is organic microplastic, while the remaining ingredients are oxides of metals and nonmetals and carbon black [37]. Thus, waste toner is considered as a suitable raw material for the synthesis of various nanomaterials. Recently, it has been used as a carbon source for the synthesis of multiwalled carbon nanotubes using a chemical vapour deposition process [50]. Xu et al. synthesized graphene oxide quantum dots from waste toner powder [51]. The synthesis was conducted using a simple hydrothermal reaction and the resulting quantum dots were applied in DNA sensing. While e-waste such as printer inks and toners are known to cause DNA damage [52], it is interesting to note that the nanoparticles generated from toners are used to detect DNA damage. In another independent study, waste toner was converted into a magnetic nanocomposite by thermal treatment followed by calcination in ammonia [41]. The nanocomposite was capable of reducing Cr (VI) in aqueous solutions and was used for the remediation of waste water.

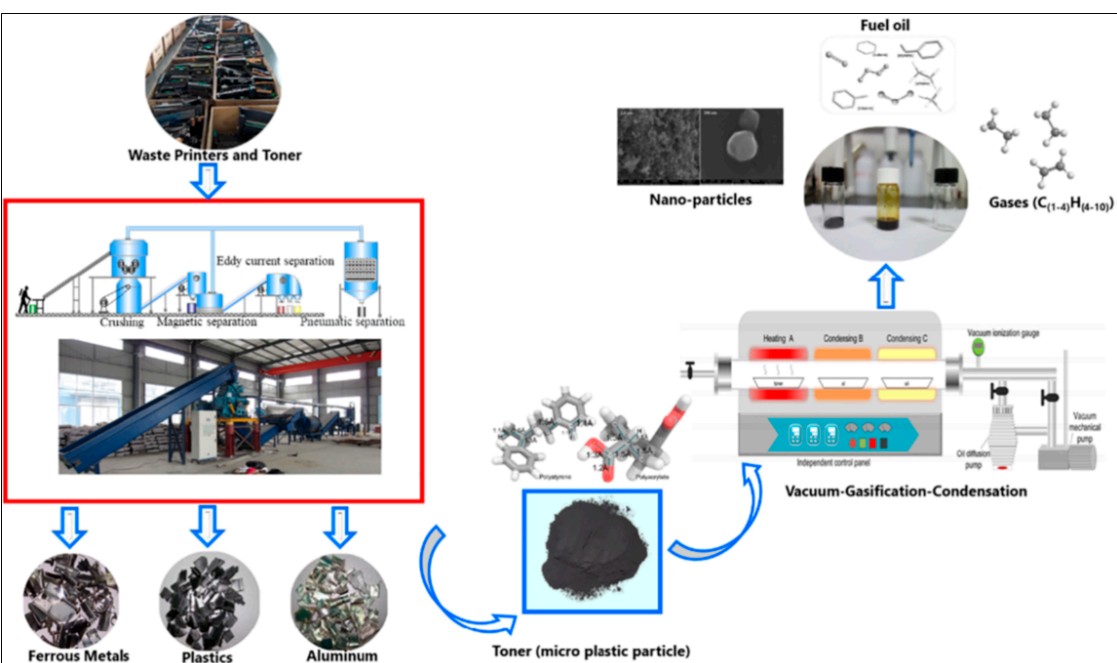

**Figure 3.** Inline processing of toner waste to generate value-added products such as nanoparticles, fuel oils, and gases (reproduced with permission from Ref. [24]).

Toner waste powder is also used as a filler to enhance the electrical conductivity of polymer blends by several orders of magnitude [42]. Because of good electrical conductivity, charging behaviour, and adsorptive capacity, toner powders are being tested as active components in power generation and storage devices. Waste toner, especially, could be more attractive for power generation devices due to lack of fusible plastic particles, which otherwise would have caused considerable resistance to electrode reactions. By eliminating the microplastic components by heat treatment, waste toner generates carbon-coated ferric oxides for application as anodes in lithium–ion batteries [43]. Kaipannan et al. reported an environmentally viable process for the thermal conversion of waste toner into carbon-coated iron oxides and applied them as active materials in supercapacitors [44]. The supercapacitors designed using the processed waste toner yielded better performance than those containing commercial mesoporous graphitized carbon black. Calcination of waste toner at 600 °C yielded magnetic $Fe_2O_3$ particles, which were used to form photocatalysts for degradation of textile dye effluents [6]. In another study, carbon–$Fe_3O_4$ composites derived by the heat treatment of waste toner were used as anode materials in sodium ion batteries [53]. Three demonstration projects were conducted on toner-modified asphalt binders in the Laredo, Houston, and Pharr districts in Texas, USA [54]. Parameters including mixing/compaction temperatures, binding performance, blending time, and storage stability were studied. Increased stiffness and lower storage life were identified as the major challenges in toner-modified asphalt.

The nanocomposite nature of waste toner renders excellent mechanical and rheological performance, when blended with construction materials. An important issue in asphalt binding for pavement construction is low temperature cracking and rutting. Toner waste has been used to solve this problem. On mixing with asphalt binders at suitable proportions, toner waste powder improved the mechanical and rheological properties of the pavement [45]. Asphalt Paving & Maintenance (APM Co., Inc., West Columbia, SC) was the first U.S. contractor to use toner waste mixed asphalt binders in pavement construction [46]. Further, the hydrophobic nature of waste toner makes the asphalt binder resistant to moisture and short-term aging [47]. In a recent study, waste toner was found to improve the rut resistance and bonding strength of asphalt binders irrespective of its proportion [55]. Toner waste in combination with calcite is also being applied in concrete

as a replacement for sand [48]. The composite concrete did not leach any hazardous components in the environment. This is a viable alternative to convert the hazardous toner waste into a nonhazardous construction material. Another interesting utilization of waste toner emerging in the current decade is in microwave absorbers. The ferric oxide component of toner powder makes it an excellent active matrix for microwave absorbers. In a recent report, waste toner powder was demonstrated to be a better active matrix then the conventional passive paraffin matrices for microwave absorbers based on MnNiZn ferrites [49].

Toner manufacturers are also taking steps towards inline recycling of toner waste mainly through two recycling loops called internal recycling and external recycling [13]. In the internal loop, fine toner particles wasted in the grinding step are screened and recycled. In the external loop, used toner received from consumers is screened and used in the manufacturing process to minimize the input of virgin raw materials. These inline recycling methods were found to minimize the wastage and usage of raw materials as per the reported data (Table 4). More design tools to include recycling and remanufacturing aspects of toners are needed for relevant lifecycle analysis in the printing industry [56].

**Table 4.** Improvements in lifecycle associated with recycling of toner waste [13].

| Waste/Material | Without Recycling (kg/mton) | With Recycling (kg/mton) | Percent Reduction |
|---|---|---|---|
| Solid waste produced in the lifecycle | 1020 | 780 | 24% |
| Virgin materials used in toner lifecycle | 2530 | 1790 | 29% |

Innovations in printing and xerography technologies are focusing more towards sustainable production and consumption, which is one of the United Nations Sustainable Development Goals (SDG # 12). It is possible to implement the SDG by adopting the requirements of a circular economy, namely, reduce, reuse, recycle, remanufacture, redesign, and repair right from the manufacturing of original toners down to the management of waste toners. Recently HP Ltd. (CA, USA), has released its progress in sustainable printing technology. The company has manufactured printers with 30% closed loop, postconsumer plastics, a toner tank with 25% recycled plastic, and a toner reload kit with 75% recycled plastic [57]. A circular economy can be achieved in the manufacture and management of toners by incorporating the twelve principles of green chemistry (Figure 4) suggested by Paul Anastas and John Warner [58]. Xerox XRCC Ltd. (Ontario, Canada) has developed an Emulsion Aggregation toner technology adopting the twelve principles of green chemistry [59]. The novel manufacturing process used 35% less energy/kg toner compared to conventional manufacturing. In addition, the novel process yielded smaller toner particles, which in turn resulted in 40% lower toner mass per print page and an amazing reduction in toner waste by 68%. Industries are currently resorting to eco-friendly toner alternatives to ensure energy economy, less carbon footprint, and environmental sustainability, without compromising print quality. Kyocera Document Solutions Ltd. (Osaka, Japan), has developed an eco-friendly toner, which fuses at lower temperatures, promising an energy saving of 30% in the printing process [60]. Further, no organic solvent was used in the toner manufacturing process, which makes it environmentally friendly. To make the eco-friendly alternatives commercially viable, their social and economic acceptance need to be improved by creating awareness and incentives. For the widely operating toner technologies in current use, the principles of green chemistry need to be integrated in the recycling process. Recently, The Centre for Green Chemistry & Green Engineering at The Yale University has developed a sustainable method to recycle rare earth metals from e-waste [61]. The formal recycling norms laid for developed countries need to be made economically viable with suitable innovations. This would limit the transboundary movement of e-waste and the subsequent environmental pollution due to informal recycling in the developing countries. Thus, a unified approach incorporating the

concepts of the circular economy and green chemistry is needed to achieve the sustainable manufacture and management of printer and copier toners.

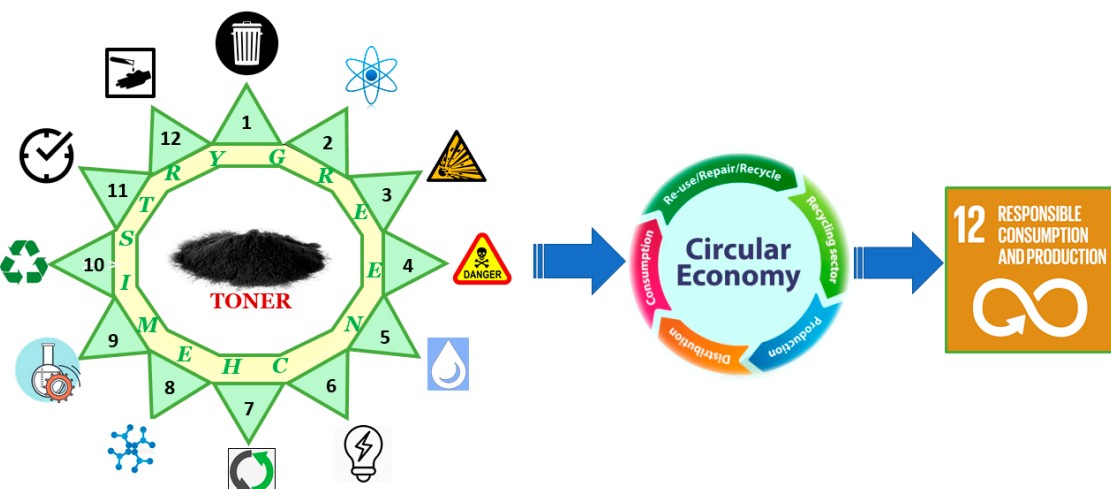

**Figure 4.** Green chemistry principles: approach to achieve a sustainable solution to the issues in the manufacture and management of toner powders.

## 5. Future Directions

With the increasing environmental concerns of waste toners piling up in landfills, researchers and governments have started looking at them as a potential threat. Efforts from individual manufacturers such as take-back programs, recycling campaigns, and alternative technologies for a greener lifecycle are observed here and there. In parallel, researchers are trying to find alternative applications for waste toners such as generation of value-added products, remanufacturing, nanomaterial synthesis, and power generation. The consumers and informal recyclers, on the other hand, are almost unaware of the environmental threats of disposed toners and the economic potential of remanufactured toners, respectively. What is lacking probably is an integration of all these efforts, channeled by government policies, legislations, benchmarks, and infrastructure to achieve a sustainable management of toner waste.

### 5.1. Policies and Legislations

Legislations based on EPR policies are needed to channel the collection, recycling, and reuse of toner waste. Different strategies might be required for developed and developing countries to implement legislation. Developed countries are capable of affording the cost, labour, and infrastructure for recycling. Yet, it is not happening in reality due to reluctance and negligence by the manufacturers and recycling agencies. Legislation is needed for the monitoring and systematic documentation of formal recycling processes in developed countries. The recycling practices might be incentivized and the byproducts/remanufactured products be exempted from taxation. Whereas in developing countries, the scenario is entirely different and the implementation of e-waste management legislation might be more challenging. Universal definitions of e-waste management are needed to create clear public awareness in developing countries. As the informal recycling sector is well-established and supports the livelihood of underprivileged societies, it is better to engage the sector while framing the legislation for developing countries. The informal sector could be formalized with norms for eco-friendly treatment and disposal of byproducts and integrated with the formal recycling sector. The Institute for Global Environment Strategies suggested a phase-in approach to implement EPR policies for e-waste management in developing countries [18]. The approach could be modified to implement the policies for toner waste management as shown in Figure 5.

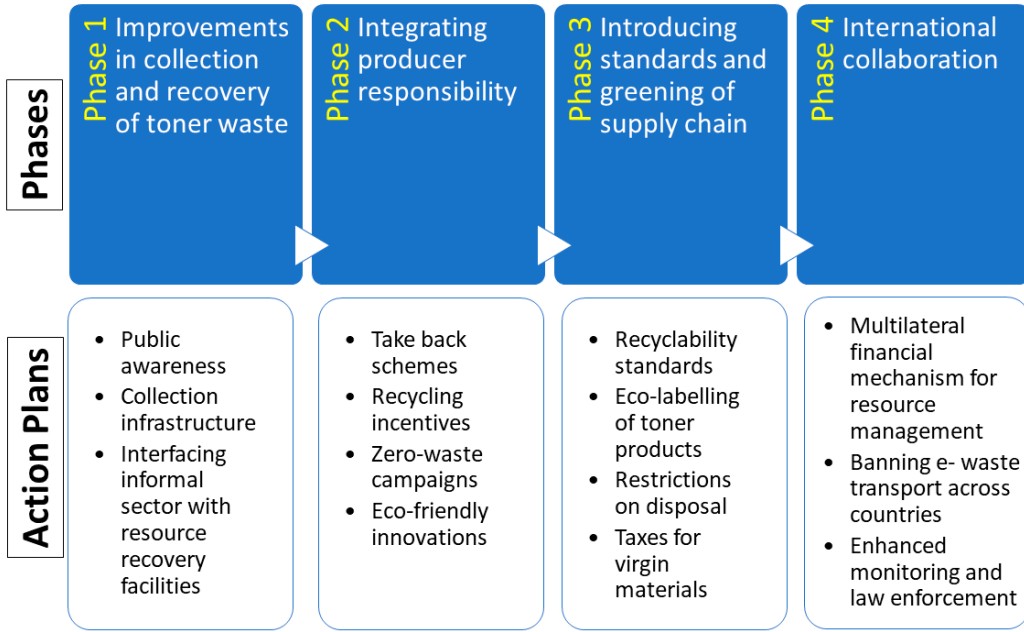

**Figure 5.** Phased approach and action plans for the strategic implementation of toner waste management based on extended producer responsibility.

### 5.2. Standards and Eco-Labelling

Toners are manufactured following the International Standard for permanence and durability, ISO 11798. This helps in maintaining the quality of toners, which are intended to be stable under different environmental conditions. On the other hand, processing of waste toners and recycling becomes difficult. There are excellent ISO standards to ensure the ecological safety of products, which could also be adopted in the manufacture and recycling of toners. For instance, ISO/TS 14067:2013 specifies guidelines for quantification and communication of the carbon footprint (CFP) of products. ISO 14040 and ISO 14044 on lifecycle assessment are to be followed for quantification and ISO 14020, ISO 14024 and ISO 14025 on environmental labels and declarations are to be followed for communication. Introducing these certifications and labels would ensure the manufacture of recyclable toners as well as educate users to choose eco-friendly toners.

### 5.3. Innovations in Recycling

A number of reports are emerging on recycling, remanufacturing, and reuse of toner waste. However, the environmental safety of these methods should be considered before adopting them on a large scale. To compare various approaches of recycling and make a choice based on environmental safety, standard protocols are needed to assess and certify the recyclability of toners just as the RecyClass™ recyclability protocols (Version 2.0, April 2021) are adopted for assessing polypropylene containers. More technological innovations are needed to recycle and reuse toner waste on a large scale as well as in an environmentally-friendly manner.

In conclusion, for effective management of toner waste, efforts are needed in different aspects including greener technologies for toner manufacture, innovative methods for toner recycling/reuse/conversion into value-added products, stringent policies for monitoring and controlling the disposal/export of toner waste, infrastructure for collection and transport of toner waste, legislations integrating the formal and informal recycling sectors, and funding mechanisms for implementation.

**Funding:** This research received no external funding.

**Conflicts of Interest:** The author declares no conflict of interest.

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
