# Peer review of "Challenges and Emerging Trends in Toner Waste Recycling: A Review"

_recycling, doi:10.3390/recycling6030057_

Round 1
Reviewer 1 Report
It is a well-written review, which clarifies the challenges and trends in waste toner recycling. In different research areas, waste toner has been utilized such as asphalt additives. Thus, I suggest the authors read the literature systematically. For example, the paper "Evaluation of workability and mechanical properties of asphalt binder and mixture modified with waste toner" has clarified the use of toner for toner. Also, other similar research should be added to this manuscript as well.Author Response
Please see the attachment.

Reviewer 2 Report
This paper offers a broad overview of toner waste as an environmental problems along with a review of research that has sought to improve the reuse and recycling of toner waste.
In general, this paper feels somewhat underwhelming. It is a very short article in terms of the written substance and content and feels more like a background section for an article on toner recycling rather than a comprehensive literature review on toner recycling. There is little specific discussion on toner recycling policies and laws and there is also little discussion on the informal sector and how toner is currently recycled (e.g., I doubt that all toner waste that isn't formally recycled ends up in the landfill, as the article suggests), moreover, there is little discussion on the challenge of collection, which has been recognized as one of the most pressing challenges to achieve e-waste recycling targets.
Below, I have provided some specific issues throughout the paper:
Page 1: It is important to distinguish that this 17.4 % is what has been collected and recycled in formal channels. There is a much higher percentage of e-waste that is collected and recycled in informal channels.
Page 1: It is unclear what you mean by this sentence: “Compared to the amount of e-waste generated per capita, the extent of documentation and recycling is not yet significant.”
Page 1: There needs to be a bit more detail and nuance on your description of e-waste legislation and driver that shape e-waste export. I would explicitly discuss Extended Producer Responsibility (EPR) as the foundational principle guiding e-waste legislation.
Page 2: Can you make more clear the linkage between multiple cartridges and serious environmental pollution?
Page 3: It is interesting to bring up the ISO standards for toners that make recycling challenging. I wonder if you come back to critique and make recommendations to the ISO standards to promote toner cartridge reusability and dismantling?
Page 3: wasted not waster
Page 3: It is not clear what there formal recycling norms are
Page 4: Figure 2 doesn't offer enough information to be worth including. It raises more questions than it answers.
Page 4: It is important to distinguish that this was from e-waste recycling generally and on toner recycling exclusively.
Page 4: Can you talk a little about the economic value or recycling toner? Is there any economic value to be recouped? Is the informal sector recycling toner and profiting from it? What processes are they using?
Page 5: can you explain what you mean by reservoir?
Page 5: Who is the "we" you are referring to?
Page 7: There is no real discussion on the collection on toners, which is one of the most challenging aspects of e-waste management.
Page 7: The 12 principles of green chemistry are not articulated in the figure.
Reviewer 3 Report
I would like to appreciate the effort put into the preparation of this article. The paper is very interesting and the issues raised are important and topical.
However, I suggest that we consider the following:
1. Completing the article with a proposed method of dealing with the recycling of e-waste. Perhaps a comparison of procedures in different countries and their advantages and disadvantages.
2) Europe is much smaller than the area shown in green in Figure 1.
3. What I miss in the overview paper is a critique of what has been done so far and a vision for further development in this area. The information presented in Future directions is also an overview of the work carried out.
4. The paper lacks a conclusion. There is only a very general figure without any comment from the authors.
5. Minor remarks are given in the paper.
I believe that the paper should be further expanded before publication. I am not asking for more publication, but for the authors' comments and thoughts in this regard. In its current form it can be said that it is only a literature review.

Author Response
Response to Reviewers’ comments
I am thankful to the reviewers and the Editor for the valuable suggestions and critical comments, which undoubtedly have given a new shape to the manuscript.
In the revised manuscript, changes are marked with “Track changes” option and shown in blue colour font.
Reviewer 3 :
I would like to appreciate the effort put into the preparation of this article. The paper is very interesting and the issues raised are important and topical.
However, I suggest that we consider the following:
- Comments: Completing the article with a proposed method of dealing with the recycling of e-waste. Perhaps a comparison of procedures indifferent countries and their advantages and disadvantages.
Response: Methods of dealing with the recycling of e-waste from different perspectives such as policies, standards, innovations and implementation are now included in the revised manuscript. Formal and informal recycling procedures in different countries are also discussed. Introduced a table comparing EPR-based legislations in different countries. I have included reference citations which describe the legislations in developed and developing countries.
Modification effected: Section 5 (future directions) has been completely rewritten based on the reviewer’s suggestion. References #5, 22, 30 are inserted and discussions included in the text. Table 1 on EPR-based legislations in developing countries is included.
- Comment: Europe is much smaller than the area shown in green in Figure 1.
Modification effected: I have modified Figure 1 with a better representation of continents.
- Comments: What I miss in the overview paper is a critique of what has been done so far and a vision for further development in this area. The information presented in Future directions is also an overview of the work carried out.
Response: I am thankful for the critical suggestion. The section on Future directions has been modified completely reflecting the present situation and recommendations. The content that was present earlier in this section has been moved to the previous section on Emerging Trends.
Modification effected: Section 5 (Future directions) has been completely rewritten and the previous content has been moved to Section 4 (Emerging trends).
- Comments: The paper lacks a conclusion. There is only a very general figure without any comment from the authors.
Response: A conclusion has been added in Future directions section. A phase-wise approach for the implementation of EPR-based toner waste recycling is presented. The figures have been modified in the light of the comments received from all reviewers and they are discussed in the text to convey relevant information.
Modification effected: Figures 1 & 4 are modified, Figure 2 is replaced, Figure 5, Tables 1 & 4 are newly inserted.
- Comments: Minor remarks are given in the paper. I believe that the paper should be further expanded before publication. I am not asking for more publication, but for the authors' comments and thoughts in this regard. In its current form it can be said that it is only a literature review.
Response to these remarks is also entered in the pdf document (please see the attachment). They are also given below for quick reference.
- Is Europe that big ?
Response: Figure 1 is redrawn to correct the error.
- Do you know the market share of these solutions?
Response: Sorry, I could not find reliable data for the market share of these solutions.
- Please indicate the source of this information
Response: This is from Ref #32 and the citation is now included near the statement
- Different font colour
Response: Font colour corrected to black
